# Synergistic impacts of global warming and thermohaline circulation collapse on amphibians

Julián A. Velasco[1], Francisco Estrada [1,2,3✉], Oscar Calderón-Bustamante [1], Didier Swingedouw [4], Carolina Ureta [1,5], Carlos Gay[1] & Dimitri Defrance[6,7]

Impacts on ecosystems and biodiversity are a prominent area of research in climate change. However, little is known about the effects of abrupt climate change and climate catastrophes on them. The probability of occurrence of such events is largely unknown but the associated risks could be large enough to influence global climate policy. Amphibians are indicators of ecosystems' health and particularly sensitive to novel climate conditions. Using state-of-the-art climate model simulations, we present a global assessment of the effects of unabated global warming and a collapse of the Atlantic meridional overturning circulation (AMOC) on the distribution of 2509 amphibian species across six biogeographical realms and extinction risk categories. Global warming impacts are severe and strongly enhanced by additional and substantial AMOC weakening, showing tipping point behavior for many amphibian species. Further declines in climatically suitable areas are projected across multiple clades, and bio-geographical regions. Species loss in regional assemblages is extensive across regions, with Neotropical, Nearctic and Palearctic regions being most affected. Results underline the need to expand existing knowledge about the consequences of climate catastrophes on human and natural systems to properly assess the risks of unabated warming and the benefits of active mitigation strategies.

[1] Centro de Ciencias de la Atmosfera, Universidad Nacional Autónoma de México, CDMX, Mexico. [2] Institute for Environmental Studies, VU Amsterdam, Amsterdam, the Netherlands. [3] Programa de Investigación en Cambio Climático, Universidad Nacional Autónoma de México, CDMX, Mexico. [4] Environnements et Paléoenvironnements Océaniques et Continentaux, CNRS, Université de Bordeaux, Pessac, France. [5] Cátedra Consejo Nacional de Ciencia y Tecnología, CDMX, Mexico. [6] ESPACE-DEV, Univ Montpellier, IRD, Univ Guyane, Univ Reunion, Univ Antilles, Univ Avignon, Maison de la Télédétection, Montpellier, Cedex, France. [7] The Climate Data Factory, Paris, France. ✉email: feporrua@atmosfera.unam.mx

An ever-growing body of evidence based on observations and projections has shown that ecosystems and biodiversity are highly sensitive to changes in climatic conditions[1–3]. The geographical ranges of many terrestrial and freshwater species have moved poleward and up in altitude over the past decades and abrupt and irreversible regional changes in these ecosystems are expected during this century for high-warming scenarios[1]. A variety of stressors related to global change have pushed current extinction rates of vertebrate species between 10 and 100 times their background rates[4]. Amphibians are at higher risk than other vertebrate groups, with about ~43% of species under threat of extinction[5,6]. Further warming of the climate system would produce severe changes in the climatic range of all species. Under a high-warming climate scenario (A1B), about 57% of plants and 34% of animals are likely to lose more than half of their present suitable distributional area by 2080, being amphibians and reptiles at highest risk[1,7]. Our study takes advantage of the quality of amphibians as useful bioindicators of ecosystems health[8–11] to offer insights about more general impacts of climate catastrophes on biodiversity.

Numerical modeling of the climate system has significantly advanced over the last decades and current state-of-the-art physical models are able to successfully reproduce a wide range of aspects of observed climate and to represent complex dynamical processes and their interactions[12]. However, thresholds and probabilities of occurrence for tipping points such as abrupt and potentially irreversible climate changes are highly uncertain and not well-understood or modeled[13–16]. These global catastrophic climate events include the shutdown of the Atlantic meridional overturning circulation (AMOC), the disintegration of West Antarctica and Greenland ice sheets, and the dieback of the Amazon rainforest[13,15–17].

The AMOC accounts for most of the global northward oceanic heat transport and a weakening or collapse of the AMOC would have large effects on the global climate[18]. Apart from a marked cooling in most of the northern hemisphere, a wide range of teleconnections would alter global precipitation patterns, strengthen storm tracks in the North Atlantic and lead to further warming in regions of the southern hemisphere[19,20]. Anthropogenic climate change could lead to the weakening of the AMOC through additional freshwater input in the North Atlantic from the melting of the Greenland ice sheet and from increases in precipitation[21] over this region as well as changes in heat fluxes[22]. Recent reconstructions suggest an unprecedented weakening of the AMOC[23], and future warming is very likely to further debilitate it[17]. Most of the climate models included in the CMIP5 experiment[24] project a moderate weakening of the AMOC during this century, although with a high degree of uncertainty[17]. However, it has been noted that current models may have common biases that favor a stable AMOC and that CMIP5 simulations do not include interactive ice sheet component[25,26]. This could lead to underestimating the risk of a large slowdown or collapse of the AMOC and hosing experiments are alternatives to explore the consequences of large changes in the AMOC due to Greenland ice sheet melting[19,20,27]. In this type of experiment, an external flux of freshwater is imposed in the North Atlantic to simulate the climate impacts of massive melting from the Greenland ice sheet[19,27], which may be justified by the very large uncertainty that concerns the fate of Greenland ice sheet melting, and in the AMOC response, due to unresolved crucial processes in present-day ice sheet models.

There is a very limited number of studies and a large uncertainty about the potential impacts of a shutdown of the AMOC on natural and human systems[28–30]. Developing adequate adaptation strategies for such a low-probability large-impact event remains a substantial challenge as highlighted in recent IPCC SROCC report[17]. Previous work on the effects of an AMOC collapse on ecosystems and biodiversity is mostly based on broad impact categories and simplified modeling approaches. The available estimates of the effects of a shutdown of the AMOC under a high-warming scenario on terrestrial net primary productivity are limited. They show strong regional differences and depend considerably on the assumptions about the $CO_2$ fertilization effect[19,31]. Decreases of net primary productivity in marine ecosystems have been associated with changes in the AMOC[32] and the effects of an AMOC shutdown could lead to the collapse of North Atlantic plankton stocks[33]. Here we present a global assessment based on an ecological niche modeling approach (see "Methods", Supplementary Information S3) of the synergistic effects of unabated global warming and different levels of AMOC slowdown for 2509 amphibian species across six biogeographical realms (Fig. S1).

## Results and discussion

The ecological niche model projections are based on five bioclimatic indices derived from monthly temperature and precipitation data from climate simulations of the Institut Pierre Simon Laplace low-resolution coupled ocean-atmosphere model (IPSL-CM5-LR; see "Methods", Supplementary Information S2). The control run is based on the RCP8.5 emissions scenario for the period 2006–2100 and four hosing experiments were superimposed to the RCP8.5 which added 0.11, 0.22, 0.34, and 0.68 Sv (1 Sv = $10^6$ m³/s; labeled as A, B, C, and D, respectively) of freshwater released in the North Atlantic from 2020 to 2070[29]. These hosing experiments lead to different levels of additional weakening of AMOC over the century (Fig. S2).

**Impacts on amphibian species under the RCP8.5 scenario**. Under a high-emissions scenario (RPC8.5), associated with a high level of warming at the global scale, the projected range contractions for amphibians varies widely across biogeographical regions (Fig. 1) but tend to be similar across extinction risk status and high-level taxonomic groupings (Figs. S18–S19). These contraction range patterns are similar between two dispersal strategies (full dispersal and no dispersal) and the medians of the range of contraction tend to increase with warming during this century (Fig. 1). The projected range contractions values were calculated including multiple sources of uncertainty explicitly (e.g. niche modeling algorithm methods, biogeographical realms, extinction risk status, and taxonomic groupings; see Supplementary Information S3). The corresponding interquartile ranges are large in all horizons, underlying the diversity in responses from individual species within categories (realms, status, family) as well as the uncertainty in modeling (see Supplementary Information S3.4). Under the control RCP8.5 simulation, areas with lower projected range contractions for amphibian species are located in temperate regions such as Palearctic and Nearctic (Fig. 1; Fig. S1). Temperate amphibians show the lowest proportion of range loss supporting previous findings[7]. Indomalayan, Afrotropical, and Neotropical regions (Fig. S1) are highly vulnerable to even small levels of warming, reaching median range losses larger than 50% already in the 2030s and of more than 75% by the end of this century. As the climate conditions in this high-emissions scenario (RCP8.5) become more extreme toward the end of the century (i.e., 2070), the median reduction in distributional range increases (Fig. 1). Regardless of their current classification, in the 2030s, species from all endangerment status would experience a median reduction of their distribution range of about 50% and, by the end of the century, the median reduction range would increase to about 75%, with the exception of critically endangered under full dispersion (Figs. S18–S19). The

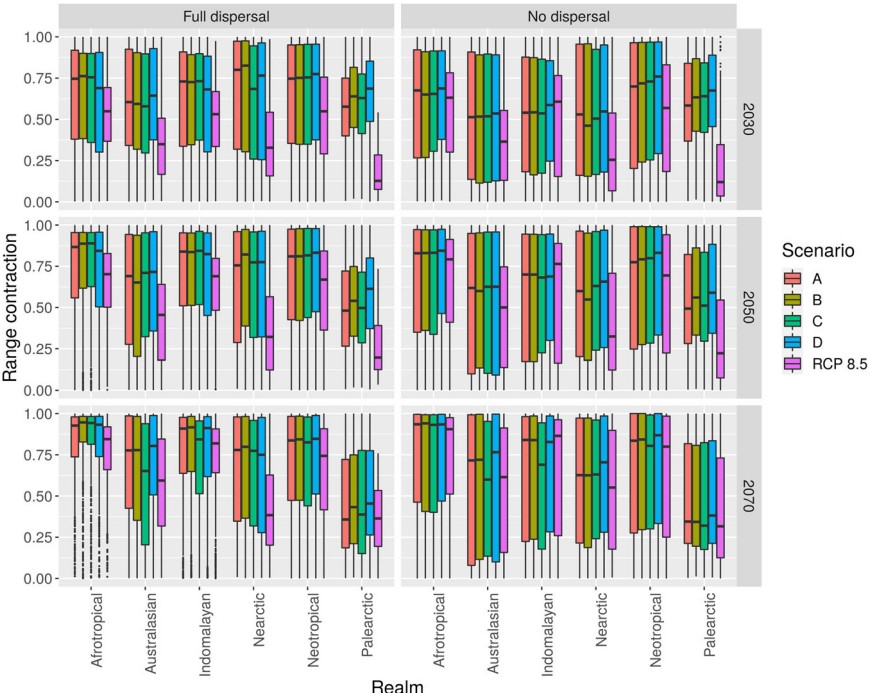

**Fig. 1 Boxplots of projected range contractions for amphibian species.** Boxplots of projected range contractions for amphibian species contracting their ranges under a high-emissions scenario (RCP8.5; reference scenario) and a high-emissions scenario with four hosing experiments adding 0.11, 0.22, 0.34, and 0.68-Sv of freshwater (1 Sv $= 10^6$ m$^3$/s) labeled as A, B, C, and D, respectively. The number of species contracting their ranges varied across scenarios, time horizon, niche modeling algorithms, and realms (955–2068; Tables S2–S3). Range contractions are expressed in proportional terms and are shown for two dispersal scenarios (full dispersal and no dispersal). Results were averaged across three ecological niche modeling algorithms (MaxEnt, BRT, CART; see "Methods") and compared across six biogeographic regions.

median of projected range loss for all time horizons and biogeographical realms was moderate in a full dispersal scenario (0.51; 95% CI: 0.50–0.52) and high in a no dispersal scenario (0.74; 95% CI: 0.74–0.75).

**Synergistic impacts from a high-warming scenario plus additional weakening of AMOC.** The RCP8.5 plus hosing simulations, which impose an additional weakening of the AMOC, entail more severe impacts to those of the control RCP8.5 scenario with no additional freshwater discharges from Greenland ice sheets. In the hosing simulations with additional weakening of the AMOC, very large range contractions occur soon after additional freshwater starts to be added and the decrease in suitable areas continues to grow during the century for most realms (e.g., Neotropical and Palearctic realms; Fig. 1). This holds for the 2070 horizon (30-year average centered in 2070) even if the discharge of additional freshwater stops at the middle of this period. Figure S2 shows that AMOC decreases about 80% (scenario D) in the 2070s and does not significantly recover during this century. Changes in climate under the hosing experiments markedly differ from those of the RCP8.5 with respect to much larger contrasts between the coldest and warmest months and precipitation changes most notably in the North Atlantic bordering regions (Figs. S8–S16; Supplementary Information S2.2, S2.3). Changes in the AMOC are persistent and their effects over climate (e.g., changes in temperature and precipitation patterns, variability, teleconnections) are still considerable decades after the freshwater forcing is removed. In consequence, the impacts on natural systems of an AMOC collapse are persistent and may extend well beyond this century. The median of the decrease in the climatic suitable range distribution for amphibian species at the global scale in the hosing experiments is larger than that of the RCP8.5 control run, under both the full

dispersal (0.67; 95% CI: 0.66–0.67) and no dispersal scenarios (0.85; 95% CI: 0.84–0.85). Moreover, the characteristics of these scenarios are informative about the risks such climate catastrophes may impose in the short-, medium- and long-horizons. Results are highly nonlinear between time horizons and across the hosing experiments ($F = 553$, $p < 0.001$; Fig. 1), the range contraction of species is much larger compared to the control scenario across biogeographical realms ($F = 407.1$, $p < 0.001$), taxonomic groupings ($F = 74.33$, $p < 0.001$; Fig. S18), extinction risk status ($F = 50.05$, $p < 0.001$; Fig. S19), but not between freshwater discharge levels ($F = 1.65$, $p = 0.176$; Fig. 1; Supplementary Information S3). The most diverse families (i.e., those with more than 100 species) exhibit different trends in range contractions under high warming and for the different AMOC weakening scenarios (Fig. S18) suggesting that abrupt climate change impacts will be pervasive across different clades, regardless of its phylogenetic position or extinction risk status (Figs. S18–19). Furthermore, even the smallest additional slowdown of the AMOC (scenario A) produces a large and rapid contraction of the distributional range of amphibians. In the 2030s, about 10 years after the hosing experiment started, the range contraction increases in all realms, status, and most families. In particular, the median contraction in the Palearctic region is about 50% higher than under the RCP8.5 control simulation, while the reduction for the Nearctic region increases by 25–50%, depending on the level of freshwater released and if dispersal is allowed or not. Additional AMOC weakening could in principle be associated with lower impacts in the Palearctic region caused by lower warming levels in high latitudes. However, under the hosing experiments, climate becomes considerably more dissimilar to current conditions than under the RCP8.5 scenario (Figs. S8–S16), which leads to higher impacts than under the RCP8.5 scenario. Paleoclimatic studies and model simulations

suggest that under an AMOC collapse winters get much colder but that summers get considerably warmer in the Palearctic region[34]; this is also shown by the climate projections in our study (Fig. S8) and other models[20]. The speed of change and the insensitivity of the results to the amount of freshwater released suggest the existence of tipping points that lead to abrupt contractions in the species range. Similar losses in the species ranges are projected for mid-century. Toward the end of the century, even if species have the possibility to track their climatic niches (full dispersal scenario), the range reductions will be more severe for most regions under hosing experiments. During the 2070s, the Palearctic region would have climatic conditions that recover some of the suitable distributional area lost for amphibians. Nonetheless, this does not imply that these species would recover after experiencing the severe effects of an AMOC disruption since such impacts may lead to irreversible changes in ecosystems[35]. The differences between hosing and control experiments are smaller if no dispersal is allowed. This result is in part due to the much larger variability in the species' response under no dispersal.

In addition to projections of changes in suitable distributional area, scenarios about loss in biodiversity (i.e., changes in number of species) help to better represent the risks an AMOC collapse could imply. The percentage of amphibian species loss varies substantially across regions, time horizons, and algorithms (Fig. 2; Figs. S23–S29; Supplementary Information S3.6). The highest percentages of species loss are expected to occur toward the end of the century (Fig. 2, Figs. S23–S32). The projected patterns suggest that the impacts of additional AMOC weakening will be strongly nonlinear across spatial and temporal dimensions, and freshwater discharge levels. Loss in species richness is dramatic for scenario D due to the highly dissimilar climate conditions this scenario imposes and to tipping point behavior produced by thresholds in species loss modeling (sections S2.3, S3.4, and S3.6). By 2070, decreases in richness could reach about 70–80% in most of the Neotropics, the southern parts of the Palearctic and Nearctic realms, as well as in south Africa, east Europe, southeast of Asia, and the Middle East (Fig. 2d). Most amphibian species are currently hosted in these areas. These results suggest that changes in regional species pools (i.e., amphibian communities) under high-emissions scenarios could be already dramatic. However, additional and substantial AMOC weakening could

produce much higher impacts and have profound consequences on amphibian assemblage composition (e.g., biotic homogenization)[36] and on other biodiversity dimensions (e.g., functional and phylogenetic measures)[37].

Altogether, these results suggest that the impact of a substantial weakening of AMOC might be extensive across many clades and biogeographical regions. Higher rates of projected range contractions and loss of amphibian species are more severe in tropical regions as Neotropics and also in temperate regions as Palearctic and Nearctic. These regions exhibit substantial anomalies in temperature and precipitation and novel climatic conditions (Figs. S8–S16). This contrasts with previous studies that have ignored climate catastrophes and which results suggest that amphibian temperate species may not be severely impacted by climate change[38]. Our study illustrates the need of conducting global, regional, and local assessments of the potential impacts of climate catastrophes that could occur under anthropogenic climate change. Results underline the importance of using different approaches (cross-species and assemblage-based approach) to investigate the complexities of climate change impacts on biodiversity across spatial and phylogenetic scales. A better knowledge of non-linearities and irreversibilities in the responses of human and natural systems to low-probability high-impact events would help better gauging the risks of global warming[39,40], and supporting the development of sounder climate policy.

## Methods
Observational datasets of temperature, precipitation, and bioclimatic indices were obtained from the WorldClim database version 2[41,42]. Monthly future projections of climate variables were supplied by the Institut Pierre Simon Laplace and correspond to their low-resolution coupled ocean-atmosphere model (IPSL-CM5-LR). Five simulations were used in this paper: a control run based on the RCP8.5 scenario 2006–2100 and; four hosing experiments, using the same RCP8.5 scenario, but on which 0.11, 0.22, 0.34, and 0.68 Sv (1 Sv = $10^6$ m$^3$/s) of freshwater released in the North Atlantic from 2020 to 2070 are superimposed (see SI section S2.1 for model and experiments descriptions). To correct for any systematic bias in the climate projections, these projections were shifted by the mean bias between the modeled and observed climatology over the period 1970–2000[29]. We calculated five bioclimatic indices to evaluate the impacts on amphibian's distributions (see below; Figs. S3–S16 and Supplementary Information S2).

We used distributional data for all amphibian species from IUCN database (5547 species; http://www.iucnredlist.org). Ranges were converted in a raster format with a pixel resolution of 0.3° (~33 km × 33 km equal-area grid cells) and 1° (~100 km × 100 km equal-area grid cells). We selected endemic species from each

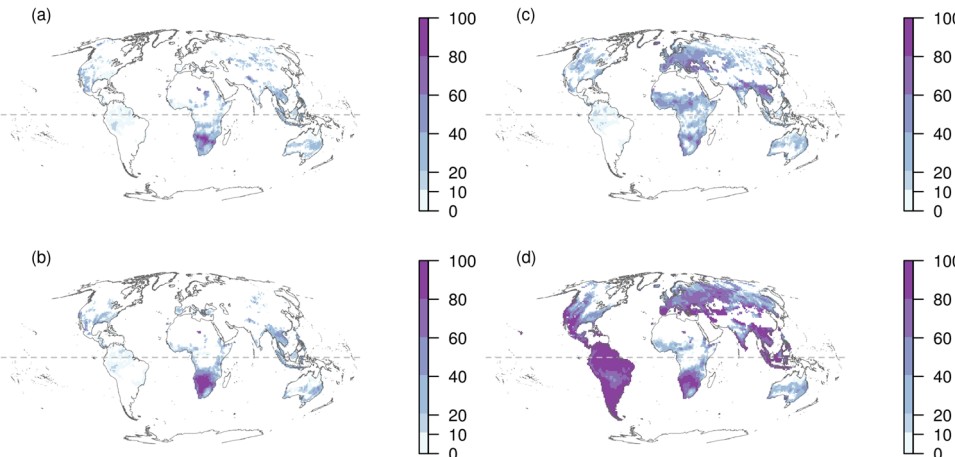

**Fig. 2 Geographical patterns of percentage of amphibian species loss.** Geographical patterns of percentage of amphibian species loss under a full dispersal scenario across six biogeographical regions (Afrotropical, Australasian, Indomalaya, Nearctic, Neotropical, and Palearctic) for different climate change scenarios. Panels **a** and **b** correspond to mapping of the percentage of species loss per pixel (1°) under a high-warming scenario (RCP8.5) for 2030 and 2070, respectively. Panels **c** and **d** correspond to mapping of the percentage of species loss under the hosing experiment D (0.68-Sv; 1 Sv = 10$^6$ m$^3$/s) for 2030 and 2070, respectively. The geographical patterns were calculated averaging the results from three niche modeling algorithm (Maxent, BRT, CART) to account for model uncertainty explicitly in the mapping of species loss across the globe.

biogeographical realm that had at least 50 pixels of presence (Table S1). The spatial resolutions (0.3° and 1°) were selected to reduce potential errors coming from these distributional sources. The same data and spatial resolution have been widely used in other macroecological studies either for evaluating impacts of climate change or for addressing ecological and/or evolutionary questions at these coarse-grain scales[43–48]. Several studies have found that distributional data either from GBIF occurrence data or IUCN expert range maps produce highly similar estimates of climatic niche conditions experienced by species[49,50]. Ficetola et al.[51] found that amphibian range maps from IUCN represent the known distributions of most amphibian species relatively well.

We focus on six realms including Afrotropical, Australasian, Indomalayan, Nearctic, Neotropical, and Palearctic (Fig. S1)[52]. Species from other realms were excluded due to low sample size (i.e., <50 presence points). We used five observed bioclimatic variables from the Worldclim database to calibrate models that have been considered important for amphibian's biology in previous studies[44,53]. These bioclimatic variables have a strong influence on species' distributions and functional traits[44,48,53,54] and have been used to model species' distributional areas both in current[44] and future climate change conditions[48,55]. The bioclimatic variables are Annual mean temperature (bio1), maximum temperature of the warmest month (bio5), minimum temperature of coldest month (bio6), annual precipitation (bio12), and precipitation seasonality (bio15). These variables represent both means (bio1, bio12), extremes (bio5 and bio6), and seasonality (bio15) conditions which jointly determine the probability of occurrence of a species in each grid cell. These variables were up-scaled from 10 km to a 33 km and 100 km resolution, respectively.

**Statistics and reproducibility.** Species distribution modeling for 2509 amphibian species (Table S1) was performed using the sdm R package[56]. We randomly partitioned the present data into two sets for model calibration (70%) and model validation (30%). We generated a set of pseudoabsences in each biogeographical region using the *ecospat.rand.pseudoabsences* function from ecospat R package[57]. The number of pseudoabsences was three times the number of presences for each species[56]. We adopt this strategy to maximize the number of pseudoabsences used for model validation using a set of standard validation metrics[58,59]. We use each biogeographical realm as the background area for all endemic species to maximize the potential available area for each species. This allows to estimate relatively well the entire set of climatic conditions that a given species likely experienced through its evolutionary and/or biogeographic history[60,61].

We selected a set of 15 species at random for each biogeographical realm and estimated species distribution models using five algorithms: MaxEnt, MARS, CART, ANN, GLM, and BRT. We explored which algorithms exhibit the best predictive performance across a set of metrics (e.g., omission rate, AUC, TSS, and Kappa; a full description of these validation metrics is available in the literature[58,59]. A full description of each model algorithm can be found in Peterson et al.[60] and Guisan et al.[62]. We selected only three algorithms (MaxEnt, CART, and BRT) which showed a relatively good model performance based on the low omission rate, high AUC, and TSS values, and a low omission rate. Then, we ran models for 2509 species using these three algorithms and ten (10) runs of random subsampling for each algorithm (Table S1). All three algorithms exhibited similar high predictive performance (Fig. S18; Supplementary Information S3). We excluded from subsequent analyses those species with poor performance denoted by low values in validation metrics (e.g., TSS ≤ 0.4).

We transferred each model to future scenarios of climate change using a control scenario based on the RCP8.5 emissions scenario for the period 2006–2100 and the four RCP8.5 plus hosing experiments described above. The output from each species distribution model was converted to binary maps (i.e., presence and absence) using three threshold criteria: minimum training presence (MTP), equal sensitivity and specificity (ESS), and maximum sensitivity and specificity (MSS)[63]. As these threshold criteria had a relatively similar impact on our estimates of projected range losses, thus we only show results for the 10thTP criteria (Fig. S21; Supplementary Information S3.4). In addition, very similar patterns were found for simulation A using 1° and 0.33° spatial resolutions (Fig. S22). All other analyses were conducted at a spatial resolution of 0.33°.

We projected impacts from climate change catastrophic scenarios on amphibian diversity using two approaches: the first based on the individual projected species response to different hosing experiments superimposed on a high-warming scenario, and the second based on the loss of species richness (i.e., diversity deficits) from regional assemblages. In the first approach, we estimated the potential suitable area (i.e., pixels predicted as presences) for each species in the current climate conditions and the different future climate change scenarios using the binary maps. The projected range losses for each species was calculated using this formula:

$$q = (p1 - p2)/p1 \qquad (1)$$

where $q$ is the proportion of gained and/or lost area; $p1$ refers to pixels predicted as presences in current potential distributions and $p2$ refers to pixels predicted as presences in future potential distributions. As we found high variability in projected range contractions among model algorithms, we averaged range contraction estimates to incorporate uncertainty explicitly. We compared the projected proportion of range contractions (i.e., range losses) between realms,

extinction risk status, and the most diverse amphibian families (i.e., those with more than 100 species; Figs. S18–S19). In general, the projected range losses were robust to the very well-known uncertainty generated from model algorithm selection (Fig. S20)[48,64]. The projected range losses were evaluated under a full dispersal and a non-dispersal scenario and we found that under a non-dispersal scenario (i.e., species lacks the ability to colonize areas outside its current distribution), the projected range contractions were more severe (Fig. 1).

In the second approach, we stacked all binary maps from endemic species of each realm and overlaid then in a grid of 1° × 1° (~12544 km²) to calculate the number of species richness for current climate conditions and for each of the future climate change scenarios (RCP8.5, A, B, C, D), and time horizon (2030, 2050, 2070). We calculated the species diversity deficit (i.e., percentage of species loss) from each of the four hosing experiments (A, B, C, D) against the current diversity (i.e., the spatial pattern of species lost in each hosing experiment; Supplementary Information S36).

The mapping of percentage of species loss varied widely across algorithms and time horizons (Figs. S23–S27). The average of species loss across niche modeling algorithms under the control scenario shows that higher diversity deficits are projected for southern Africa (>80% in species loss; Fig. S28). Under this control scenario, percentage of species loss was relatively similar across the three-time horizons (Fig. S28). In contrast, the averages of species loss across niche algorithms under the hosing experiments show that results are relatively similar across freshwater discharge levels during the first years of the experiment (Fig. S29). However, at the end of the hosing experiment (2070) and with higher levels of freshwater discharge, the projected percentage of species loss tends to be more widespread and severe (>80%; Figs. 2, S29, S30–S32). This suggests that the projected percentage of species loss across geography are highly non-stationary across spatial and temporal dimensions, and freshwater discharge scales.

Although we explicitly incorporate multiple uncertainty sources in our estimates of range contractions (Supplementary Information S3), we call for further studies at fine-grain scales with high-quality data (e.g., GBIF datasets). In addition, the generality of these results needs to be determined across other taxa and complementary biodiversity facets beyond taxonomic diversity (e.g., functional and phylogenetic diversity). These future studies are necessary to promote conservation policies at the global and regional scales which account for the possibility of future climate catastrophes.

**Reporting summary.** Further information on research design is available in the Nature Research Reporting Summary linked to this article.

## Data availability
The distributional data for amphibian species is available from the IUCN database (http://www.iucnredlist.org). The data about observed climatology are available from Worldclim database (http://www.worldclim.org). The data that support the findings of this study[65] are available in Figshare with the identifier https://doi.org/10.6084/m9.figshare.13280951.v1.

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

## Acknowledgements

J.A.V. acknowledges financial support from DGAPA/PAPIIT under grant number IA201320. For the climate simulations, this work was supported by the French Atomic Commission (CEA) within the framework of the Variations Abruptes du Climat: Conséquences et Impacts éNergétiques project funded by the Département des sciences de la matière (DSM) with the DSM-Energie Program. It benefited from the high performance computing (HPC) resources made available by Grand Equipement National de Calcul Intensif, CEA, and Center National de la Recherche Scientifique.

## Author contributions

J.A.V. and F.E. contributed equally to the conceptual design; J.A.V., F.E., O.C.B., D.S., D.D., and C.U. analyzed data. J.A.V. and F.E. wrote the paper and D.S., O.C.B., C.G., and C.U. contributed to it. All authors discussed the results and commented on the manuscript.

## Competing interests

The authors declare no competing interests.
