## [Peer Review File · Communications Biology]

Reviewers' comments:

Reviewer #1 (Remarks to the Author):

This paper analyses the impacts of different levels of future AMOC slowdown on several amphibian species around the globe. The work is performed using model experiments of future climate change scenarios to force a niche model. Uncertainties relative to regional realms, extinction risk, modelling algorithm and ice sheet melting are taken into account.

The paper is well written and presented. The novelty of this study is that it uses a more complex model and updated data to evaluate the rate of extinction of amphibians. The experiments and uncertainties performed sound reasonable to investigate the posed problem. It shows that the AMOC slowdown that can be triggered by climate change is a major contributor for the long-term loss of the amphibian population.

From my expertise related to the physical part of the manuscript, my review below has to main comments: 1) This paper is missing a comprehensive analysis of the local predictors and interpretable environmental information regarding each climate scenario; and 2) Some of the main results for 2070 may point to some interannual or decadal variability on top of the associated trends.

In summary, I believe this paper has great potential, and it is a novel contribution to the scientific field for the complexity of the analysis and relationship with climate uncertainties. However, it lacks discussion about mechanisms and forcings of the modeled changes. I believe the improvements can be done in a timely manner during the revision process. Therefore, I recommend major revisions before its acceptance.

Please see below my point-by-point review:

Major comments:

1) The analysis that is missing in the manuscript is to define the forcings of the changes and mechanisms through which the populations are affected. More specifically, which of the three predictors (maximum temperature, precipitation or temperature variance) dominates the response for each scenario, and how the AMOC weakening impacts these indices regionally.

The attribution is important because some of the text claims (i.e., L. 147) that rapid Ice sheet melting impacts amphibians is not a direct relationship, but rather through a feedback mechanism on temperature and precipitation that would ultimately serve as predictions if model structural uncertainties were taken into account. Therefore, I suggest including some maps of the predictor changes or the strongest predictor for each location and scenario instead of Figures S11 and S12, which have a low quality and are harder to interpret.

2) It appears that Figure S12 shows that the deltas in the climate scenarios would favor lower temperatures and higher precipitation rates for the hosing experiments. From these plots, my guess is that the runs with stronger AMOC changes would favor the maintenance of amphibians habitats, rather than be detrimental for them. However, this is not what the model results are pointing at. In the introduction, for example, the authors state correctly (L.55-59) that the AMOC collapse produces a marked cooling in most of the northern hemisphere and a warming in the southern hemisphere.

Could the results be discussed in the light of these assumptions?

3) It is staggering the difference of species loss, particularly in South America, in 2070 for the 0.68Sv hosing scenario. This strong difference seems to be associated not only to the AMOC slowdown from the hosing experiment, but also from some interannual to decadal variability in that period, as observed in the light blue AMOC timeseries of Figure 1. That is the only period that the AMOC strength reached negative values. If this is really the case, the interannual-to-decadal variability has to be disentangled from the trends, and some analysis discussion should be included in the manuscript. The South American climate is very sensitive to the sea surface temperature

patterns in the South Atlantic, the El Niño variability, and also by the AMOC changes driven by atmospheric variability such as the NAO (Parsons et al., 2013; <https://doi.org/10.1002/2013GL058454>). Therefore, attribution of causes of species losses has to be made carefully, and these effects taken into account.

Minor comments:

L. 52 "meridional"

L. 63 This sentence is a little misleading, The reference 20 states about a weakening in the 70s due to the Great Salinity Anomaly, which was caused by coupled atmosphere/ocean dynamics in the Beaufort gyre, and caused an unprecedented AMOC anomaly in the 20th century only. Paleoproxy data shows that the AMOC may have not only weakened but collapsed during past deglacial events. Therefore, this statement has to be revised.

L. 67 Another reference could be added here on the CMIP6 models: Jourdain, N. C., Asay-Davis, X., Hattermann, T., Straneo, F., Seroussi, H., Little, C. M., and Nowicki, S.: A protocol for calculating basal melt rates in the ISMIP6 Antarctic ice sheet projections, *The Cryosphere Discuss.*, <https://doi.org/10.5194/tc-2019-277>, in review, 2019.

L. 83 This sentence is disconnected and a little too general. The effects here are on NPP? Please specify.

L. 87 Could the authors state here how many bioclimatic indices are taken into account?

L.96-97 From the box plots, it seems that there is not much variability from the extinction risk status either.

L.98 The variance between full dispersal is much lower than from no dispersal. Is there an explanation for this?

L. 106 Replace "as" by "such as".

L.107-108 What is the explanation for the temperate regions to have lower loss?

L. 318 Why is the BRT algorithm not listed here?

L. 324 Is it the lowest omission rate or a low omission rate?

Figures:

Figure 2 - IPSL instead of ISPL

Figure S11 - Missing numbers on the right colorbar.

Figures S12 and S13 - These figures have low resolution. The lines and legend are very fuzzy. Colors for the Neotropical are different in the two figures.

Figure S15 - I suggest changing the colorbar. Since the negative values are the ones important, and most of the values are negative, the colorbar range could cover only the negative range, and white for positives.

Reviewer #2 (Remarks to the Author):

Thank you for sending me the paper "Synergistic impacts of global warming and thermohaline circulation collapse on amphibians". Since I am a climate scientist, and know little about ecology etc, I can only provide a knowledgeable review of the parts of the paper dealing with the physical

climate. I have outlined my comments below and would recommend major revisions. While this study seems novel to me, it does seem to be a niche subject (both in terms of climate change, and ecology) and I wonder about whether it has sufficiently wide appeal. However I am not familiar with Nature Biology so think that that decision should be made by the editor.

My greatest criticism is that there is little discussion of how and why the changes impact amphibians. There is a mention in the methods about 5 bioclimatic variables, however there is a mention of 19 in the extended results. I assume the other 14 were not found to be important or something? There are some figures shown of two (annual mean temperature and precip) changes shown in the extended results, however the figure are so small and grainy that I can't make them out. Do you find that these two are the most important? The RCP8.5 and hosing scenarios can have different impacts on temperature and precipitation (ie RCP8.5 causes warming but AMOC weakening cools Europe, and effects on precipitation are different in different regions). I would like to see some figures and discussion linking the ecological impacts in different regions to the physical changes seen in those regions. For instance I would expect less warming in the Arctic when the AMOC weakens (assuming that the hosing runs also include increasing GHGs) – does that mean less of an impact on the amphibians? Or are other factors more important?

Model description: Please add some more detail about the hosing experiments. From looking at the results I assume that the hosing experiments also have the time varying RCP forcing. Hence the effects of hosing are the differences between the hosing experiments and the RCP8.5 experiment. Please clarify. Also please add somewhere where the hosing is added and whether any compensation is applied.

L23-24. The authors talk about impacts being 'enhanced by AMOC weakening'. There are a couple of points here. Firstly, the RCP8.5 scenario causes AMOC weakening as well as the hosing scenarios, so here and elsewhere I think you mean 'additional AMOC weakening'. Also, the impacts on climate from an AMOC weakening alone are quite different from those from the increased greenhouse gases (GHG) in RCP8.5. Some impacts are of the opposite sign – ie increased GHG causes warming, but weakening the AMOC through hosing causes cooling over particular regions. L27 and elsewhere. As someone outside the field, the terms for different regions are confusing and not defined. If these are common terms in this journal and for the readers then ignore this comment

L55-59. References used here are not really appropriate for the points made. Ref #18 is not looking at AMOC collapse and Ref #19 is not about impacts of a collapse at all. I suggest to refer to Ref 17, 24 and Jackson et al 2015

L60-62. The main driver of AMOC weakening in future projections is actually changes in surface heat fluxes. See Gregory et al 2005. However there is large uncertainty about the

L63. Reference #20 does not show this. Reference #17 (SROCC) is an alternative

L64. Reference #22 doesn't seem to be used anywhere

L67. Ref #23 does not show this. Ref #17 is an alternative

L73-74. And the large uncertainty in AMOC response

L91. Is there a reason for these levels of hosing rather than more round numbers?

L139-140. What do you mean by tipping points here? In what? Why do you think there are tipping points?

Fig 1. I don't think this figure is particularly interesting or necessary so I think it should be included in the supplementary material. As a climate scientist, its unsurprising to see the AMOC weakening in all experiments, and I assume for biologists/ecologists, this doesn't mean a lot. The RCP8.5 scenario has increasing GHG (and the others may do too) so this has many impacts separate from the AMOC weakening. Instead it would be better to show the physical impacts that are affecting the ecology/biology. In particular I'd like to see more about the biometric indices used. Possibly a map for each (of the most significant ones) at the end of the RCP8.5 scenario and the largest hosing scenario.

Fig 2. This is a horrible figure – it's really difficult to see whether there are any patterns and needs simplifying. I suggest that if there is no statistical difference across scenario or across family/realm/status then you don't show it all and note that you have looked in the text. That way you can concentrate on where there are differences. A clearer way of showing this data might be to have eg for each realm showing timeseries (three points with range) for each scenario. That way it is easier to compare the changes over time and scenario.

Fig 3. This is an interesting figure, but little discussed. Doing something like this for the biometric variables (see comments above) would help you investigate why there was a much greater loss in

S Africa and why in the hosing experiments there was more loss in Europe and S/mid America. Please could you add something on one of the figures to show where the 6 biogeographical regions are for a wider audience.

Suppl L39. What are you normalising by for Euclidean distances? If you're using the average value in the control then that is a problem since values of some variables vary little compared to their absolute value (ie distance from zero) – for instance using temperatures in degrees C and Kelvin would give very different answers

Fig S9/11. The changes in S America are very large but only occur in one of the hosing experiments at the last time period, with no smaller changes in other experiments. This seems suspicious and needs investigating. Why is the change so large and not seen otherwise?

Fig S12. This is way too small and when I zoom in is very grainy. I can't make out the different colours as well, probably because of the resolution. I suggest using less panels.

Gregory, J. M., Dixon, K. W., Stouffer, R. J., Weaver, A. J., Driesschaert, E., Eby, M., Fichefet, T., Hasumi, H., Hu, A., Jungclaus, J. H., Kamenkovich, I. V., Levermann, A., Montoya, M., Murakami, S., Nawrath, S., Oka, A., Sokolov, A. P. & Thorpe, R. B. (2005). A model intercomparison of changes in the Atlantic thermohaline circulation in response to increasing atmospheric CO₂ concentration.. *Geophysical Research Letters*, 32. doi: 10.1029/2005GL023209

Jackson, L.C., Kahana, R., Graham, T. et al. Global and European climate impacts of a slowdown of the AMOC in a high resolution GCM. *Clim Dyn* 45, 3299–3316 (2015).
<https://doi.org/10.1007/s00382-015-2540-2>

Reviewer #3 (Remarks to the Author):

This is an interesting manuscript that used ecological niche modeling approaches to evaluate the effects of global warming and a potential collapse of the Atlantic meridional overturning circulation (AMOC) on the distribution of amphibian species across biogeographical realms and extinction risk categories. The effect of AMOC was implanted based on hosing experiments simulating the consequences of large changes in the AMOC due to Greenland ice sheet melting. I found this to be an interesting approach but have a few concerns that the authors could easily answer.

Comment 1. Line 25 (Abstract), Line 132, and Line 162 – You described that large declines are projected for life-history traits. However, I did not find any information about life-history traits in the manuscript. Are you considering the IUCN extinction risk categories as life-history traits? If so, I would not recommend and suggest rewrite.

Comment 2. Lines 95-97 – Looking at the Figure 2B, I consider that under RPC8.5 scenario the projected range contractions are similar across extinction risk status. The distribution pattern of the medians is very similar to that of the high level taxonomic groupings. What criteria/analyses are you using to say that the projected range contractions for amphibians varies widely across biogeographical regions and extinction risk status but tend to be similar across high level taxonomic groupings?

Comment 3. Lines 113-115 – I suggest that this sentence should be better elaborated. It is well-known that climate conditions under 2070 scenario will show more extreme climate conditions compared to previous ones. Considering that you are not taking into account the ability of extant species to adapt to ongoing climate change. It is not surprise that you find that range contractions tends to be higher toward 2070.

Comment 4. Lines 118-119 – Why the climate change scenario simulating a weakening of the AMOC reveal a contrasting pattern to that of the control RCP8.5 scenario? Looking at the results, my understanding is that range contractions are larger for simulations of AMOC than the control RCP8.5 scenario, but they are not contrasting pattern. Furthermore, you wrote (lines 125-126): “the range contraction of species is much larger compared to the control scenario across biogeographical realms ($F=407.1$, 127 $p<0.001$; Figure 2a), extinction risk status ($F=50.05$,

$p < 0.001$; Figure 2b), taxonomic groupings ($F=74.33$, $p < 0.001$; Figure 2c), but not between freshwater discharge levels ($F=1.65$, $p=0.176$; Figure 2a-c)". I do not know if you are writing about the three time horizons (2030, 2050, and 2070) or just 2030. Looking at the results (Figure 2a-c), in 2070 the simulations of AMOC and the control RCP8.5 scenario have similar range contractions for biogeographical realms, extinction risk status, and taxonomic groups.

Comment 5, lines 166-168. This sentence is strange. Furthermore, the paper you are citing (Araujo et al. 2006, Journal of Biogeography) did not evaluate Neotropical region. I am curious because the Nearctic and Palearctic regions have the lowest values for range contraction (Figure 2a), but have one of the highest species loss. You should explore these patterns between range contractions and species loss.

Comment 6, Line 449, Figure 1 – Use "Atlantic meridional overturning circulation (AMOC) index at 26°N" instead of "AMOC index at 26°N". Why are you not showing superimposed hosing experiments that add 0.34-Sv of freshwater? Based on the legend I can't tell what the wider and thin lines represent.

Comment 7, Line 296-297 and Figure S1 – Did you project range contractions using resolution of 1° only for AMOC scenario A? What about other three scenarios?

Reply to Reviewer #1

Note: Original comments of the reviewer are included in italics, which is followed by our response in normal letter type.

This paper analyses the impacts of different levels of future AMOC slowdown on several amphibian species around the globe. The work is performed using model experiments of future climate change scenarios to force a niche model. Uncertainties relative to regional realms, extinction risk, modelling algorithm and ice sheet melting are taken into account.

The paper is well written and presented. The novelty of this study is that it uses a more complex model and updated data to evaluate the rate of extinction of amphibians. The experiments and uncertainties performed sound reasonable to investigate the posed problem. It shows that the AMOC slowdown that can be triggered by climate change is a major contributor for the long-term loss of the amphibian population.

From my expertise related to the physical part of the manuscript, my review below has to main comments: 1) This paper is missing a comprehensive analysis of the local predictors and interpretable environmental information regarding each climate scenario; and 2) Some of the main results for 2070 may point to some interannual or decadal variability on top of the associated trends.

In summary, I believe this paper has great potential, and it is a novel contribution to the scientific field for the complexity of the analysis and relationship with climate uncertainties. However, it lacks discussion about mechanisms and forcings of the modeled changes. I believe the improvements can be done in a timely manner during the revision process. Therefore, I recommend major revisions before its acceptance.

We thank Reviewer #1 for her/his positive evaluation of our manuscript and for the helpful comments and suggestions she/he provided to improve it.

Major comments

1) The analysis that is missing in the manuscript is to define the forcings of the changes and mechanisms through which the populations are affected. More specifically, which of the three predictors (maximum temperature, precipitation or temperature variance) dominates the response for each scenario, and how the AMOC weakening impacts these indices regionally. The attribution is important because some of the text claims (i.e., L. 147) that rapid Ice sheet melting impacts amphibians is not a direct relationship, but rather through a feedback mechanism on temperature and precipitation that would ultimately serve as predictions if model structural uncertainties were taken into account. Therefore, I suggest including some maps of the predictor changes or the strongest predictor for each location and scenario instead of Figures S11 and S12, which have a low quality and are harder to interpret.

We appreciate this comment from the reviewer. We have addressed this comment by a) extending the description of the bioclimatic variables b) providing literature and adding a short discussion of the bioclimatic variables used concerning their mechanistic links to amphibian species; c) improving and extending the discussion and the graphical representation of the changes in

climate as represented by individual bioclimatic variables, as well as by multivariate Euclidean distances; d) including a discussion in the SI about the idiosyncratic responses of amphibian species to changes in climate and the large spatial heterogeneity of changes in climate in the hosing experiments. We have also reorganized the SI to improve the presentation of results and methods. These improvements are briefly described below.

The Supplementary Information now includes a more complete explanation about the set of bioclimatic indices used in our manuscript and presents a more thorough analysis and comparison of the projections of such indices under the control and hosing experiments (see SI section 2). We also include a discussion about the mechanistic links of these variables with demographic processes at local and regional scales for amphibian's species. For instance, we underline that the selected bioclimatic variables have a strong influence on species' distributions and functional traits (Araújo et al. 2008; Munguía et al. 2012; Ochoa-Ochoa et al. 2019; Thuiller et al. 2019) and that these bioclimatic variables have been used to model species' distributional areas both under current (Munguía et al. 2012) and future climate change conditions (Thuiller et al. 2019; Oliveira et al. 2020). The bioclimatic variables used in our analysis represent mean climatic conditions (bio1, bio12), extremes (bio5 and bio6), and seasonality (bio15). In the estimated niche models, these variables jointly determine the probability of occurrence of a species in each grid cell. We have added this discussion to the Methods section of the main text to provide a better explanation of these bioclimatic variables and their importance for amphibian species.

The ecological niche modeling methods used to generate probabilistic scenarios about the presence-absence and the percentage of species loss are well-known and established methods to evaluate climate change impacts at coarse-grain scales (Peterson et al. 2011; Guisan et al. 2017). The algorithms used in our manuscript are based on machine learning and multivariate regression techniques to model complex environment-species relationships. Variable selection in these methods reflects the idiosyncratic response of the different species and will depend on factors such as the species' prevalence (i.e., how much a species occupies a region), extension of the area of study, and spatial auto-correlation between predictors. Accordingly, the strongest predictors for one species are frequently not the same for another even within the same region. The diversity of the changes in climate at the grid cell level and of the idiosyncratic responses of the 2,509 amphibian species modelled makes it difficult to extract general conclusions about what combinations of the bioclimatic variables drive the overall impacts in each realm. To illustrate this point, we can consider Figure R1 in which the relative importance of the bioclimatic variables is shown for 10 species in the Afrotropical realm. This example shows the large heterogeneity in variable selection and in the relative importance of each of them for a small sample of the species considered in our analysis. Moreover, the resulting impacts of the combination of different changes in bioclimatic values on amphibian diversity are often nonlinear, which may severely hamper any possibility of a clear attribution of which drivers play the larger role in amphibian response (such attributions are usually based on linear approaches). We believe that mapping the association between Euclidean distances and percent of species loss can be more informative than individual variable importance and, for this reason, we included Figure S30 in section 3.6 of the SI. This figure illustrates the spatial heterogeneity of changes in climate and of impacts on amphibian species, as well as their association at the grid scale for scenario D (addition of 0.68 Sv of freshwater) and for a control scenario (RCP 8.5). In

most cases, large changes in climate are associated with high percentages of species loss. However, due to the diversity of species and their sensitivity, in some cases large (small) losses can occur for relatively small (large) changes in climate. This figure allows us to identify the regions and grid cells in which large changes in climate produce high losses in amphibian diversity and also where the species are more (less) sensitive to changes in climate.

Fig. R1: Relative importance (in percentage) of different explaining variables for 10 different amphibian species.

We have included new figures to show how the freshwater discharge experiment affects the variables used to generate the individual ecological niche models. These figures illustrate the degree of climatic departure from baseline conditions for each of the five bioclimatic variables used in the ecological niche models (Figure S8-S16). The magnitude of these departures is context-dependent and some regions will be more affected by temperature whereas other by changes in rainfall regimes. The delta values from these figures represent the differences between a given future scenario (e.g., RCP8.5 or RCP8.5+hosing of 0.68 Sv) and baseline conditions (1970-2000). The non-stationarity across time and space reveals the complexity of the effects a potential AMOC collapse can have on climate at the regional scales. For instance, the largest departures are observed in precipitation variables (AP and PS) across all six realms (Figure S8-S14). The departure of extreme temperatures (CMT and WMT) from baseline conditions is highly geographically-dependent and in some regions is larger (e.g., Palearctic) than in others (e.g., Neotropical).

We also re-calculated the Euclidean distances based on these five variables as a multivariate index of climatic departure. These results are included in the Supplementary Information section

S2 and are illustrated using boxplots and maps of the variation across scenarios and regions. (Figures S9-S16). The multivariate Euclidean distances reveal that some regions exhibit larger magnitudes in climate departure (e.g., Neotropical) whereas others (e.g., Palearctic) exhibit smaller. Larger departures occur for freshwater scenarios in comparison with the RCP8.5-only control scenario. The large variability in the response to the Greenland melting scenarios is a direct consequence of non-stationary processes and therefore the impacts on the species will be different in each region. In addition, these results reinforce the argument in the main text that the combination of changes in climate variables, as well as the existence of non-linearities and tipping points are very important for determining the impacts of a potential AMOC collapse on ecosystems and biodiversity.

2) It appears that Figure S12 shows that the deltas in the climate scenarios would favor lower temperatures and higher precipitation rates for the hosing experiments. From these plots, my guess is that the runs with stronger AMOC changes would favor the maintenance of amphibians habitats, rather than be detrimental for them. However, this is not what the model results are pointing at. In the introduction, for example, the authors state correctly (L.55-59) that the AMOC collapse produces a marked cooling in most of the northern hemisphere and a warming in the southern hemisphere. Could the results be discussed in the light of these assumptions?

To address the reviewer's comment, we have added additional discussion to the main text as well as improved the SI which now includes a section that focuses on the changes in bioclimatic variables under the control and hosing experiments.

The reviewer is correct in that additional AMOC weakening could in principle be associated to lower impacts in the Palearctic region caused by lower warming levels in high latitudes. However, under the hosing experiments climate becomes considerably more dissimilar to current conditions than under the RCP8.5-only scenario (Figure S16;S8) concerning the seasonal cycle, which leads to higher impacts (much larger contrasts between coldest and warmest months, large changes in precipitation). Paleoclimatic studies and model simulations suggest that under an AMOC collapse winters get much colder but that summers get considerably warmer in the Palearctic region (Schenk et al. 2018); this is also shown by the climate projections in our study (Figure S8) and also suggested in classical hosing experiments (Jackson et al. 2015). Due to these larger contrasts between the coldest and warmest months, large changes in precipitation as well as to an overall more dissimilar climate, the niche models project higher impacts for amphibian species under the hosing experiments in the Palearctic region. These climatic departures are illustrated in the new Figures S8-S16 that were added to the SI section 2.

Moreover, as mentioned in our reply to the previous comment, the effects of a disruption in AMOC over precipitation and temperature shows high spatial variability, and this variability is imparted to the impacts on amphibian species. Our results reveal that amphibian range contractions (i.e., the loss of suitable habitat) under scenarios with hosing experiments are highly variable across the world. As such, the effects of a disruption in AMOC on amphibians are mixed within some regions and species benefiting from it while others showing sharp losses in suitable area, when compared with the RCP8.5 control scenario. Figure 1 in the main text (Figure 2 in the previous version) shows this variation across biogeographical realms. For instance, under the RCP8.5 scenario amphibians from the Palearctic region tend to exhibit smaller range contractions in comparison with amphibians from the Neotropical region. The same pattern can

be observed under the hosing experiments (Figure 1, panels A-D) for Palearctic amphibian species in comparison with Neotropical species. These results indicate that range contractions are highly variable among regions and species and suggest that some species will be more affected than others. The predicted cooling in the high latitudes of the northern hemisphere (Palearctic realm) for the AMOC collapse is apparent for the coldest month temperatures (CMT, see Figure S8, S14; 2030 vs. 2070). In tropical regions, such as the Neotropical and Afrotropical realms, large changes are also projected: the drop in minimum extreme temperatures is still large and the increase in warmest month temperatures tends to be notable (Figures S8, S12 and S13). We have included a more detailed discussion in the main text and in the SI about these issues.

These climate changes are captured by our ecological niche models as all species with Arctic distributions (*Salamandrella keyserlingii*, *Rana temporaria*, *Rana arvalis* and *Rana amurensis*) suffer notable range contractions under scenarios simulating an AMOC collapse (see Figure R2 below for two examples).

Figure R2. Projections of potential distribution under climate change scenarios in 2030 for two Palearctic frog species (*Rana temporaria* and *Rana arvalis*). a) Potential distribution (in red) for *Rana temporaria* under the RCP 8.5 control scenario and b) hosing experiment of 0.68 Sv in year. c) Potential distribution (in red) for *Rana arvalis* under the RCP 8.5 control scenario and d) hosing experiment of 0.68 Sv .

3) It is staggering the difference of species loss, particularly in South America, in 2070 for the 0.68Sv hosing scenario. This strong difference seems to be associated not only to the AMOC slowdown from the hosing experiment, but also from some interannual to decadal variability in that period, as observed in the light blue AMOC timeseries of Figure 1. That is the only period that the AMOC strength reached negative values. If this is really the case, the interannual-to-decadal variability has to be disentangled from the trends, and some analysis discussion should be included in the manuscript. The South American climate is very sensitive to the sea surface temperature patterns in the South Atlantic, the El Nino variability, and also by the AMOC changes driven by atmospheric variability such as the NAO (Parsons et al., 2013;

<https://doi.org/10.1002/2013GL058454>). Therefore, attribution of causes of species losses has to be made carefully, and these effects taken into account.

We appreciate this comment from the reviewer. The climate scenarios used in our analysis are constructed using 30-year averages of GCM output centered around 2030, 2050 and 2070. This averaging procedure serves as a low-pass filter which smooths out interannual variability and allows to approximate the underlying climate conditions (Arguez and Vose 2010). Although this procedure does not completely filter out the effects climate variability (e.g., multi-decadal variability), it provides a reasonable approximation and it is the standard practice to investigate the effects of future changes in climate on human and natural systems (IPCC 2014; Anthoff et al. 2016; Defrance et al. 2017). The effects of ENSO and NAO are expected to be filtered out by the 30-year averaging procedure since they usually mainly affect interannual timescale. Other internal models like the AMV or IPV might play a role though. To fully disentangle the potential imprints of internal variability vs. forced changes, a large ensemble might be necessary, which is not available though some other methods for filtering out most of the model's internal variability have been proposed (Deser et al. 2014). However, these methods require the availability of ensembles of the same model with slightly different initial conditions and the same forcing, and such ensembles are not available yet for our AMOC experiments. Moreover, under significant melting of Greenland, multidecadal variability modes are expected to be affected and strongly damped and such changes are part of the impacts due to substantial AMOC weakening or collapse (Dong and Sutton 2007; Rocha et al. 2018; Collins and Sutherland 2019). As such, we believe that in this case the separation of low-frequency climate variability and climate signal impacts may be largely artificial. Nevertheless, we agree that internal variability may somehow blur part of the signal we are trying to isolate and we acknowledge this in the revised manuscript. The large differences in species loss in the 0.68 Sv hosing experiment as compared to control simulation around 2070 are mainly due to two factors: 1) this hosing scenario and time-horizon produce the most dissimilar climate to current conditions in the Neotropical region. This is shown in Figure S15, in which the interquartile range is wider and farther away from the control scenario than any other hosing experiments and time-horizon. This is reinforced by Figures S8 and S16; 2) in contrast to the calculation of changes in suitable areas (range contraction/expansion; Figure 1), the calculation of species loss is more complicated and involves the use of thresholds that allow going from percentage of range contraction (continuous number) to species loss (binary number indicating presence or absence). Exceeding these thresholds induce highly nonlinear (crossing of potential tipping points) behavior as is illustrated by Figure 2. Our analysis includes the uncertainty of threshold selection which indicates that results are robust to different threshold selection criteria. The output from each species distribution model was converted to binary maps (i.e., presence and absence) using three threshold criteria: minimum training presence (MTP), equal sensitivity and specificity (ESS) and maximum sensitivity and specificity (MSS). As these threshold criteria had a relatively similar impact on our estimates of projected range losses, we only show results using MTP criteria hereinafter (Figure S21; Supplementary Information S3.4). This is discussed in the Methods section of the main text and in the SI section 3.4.

Our analysis was set up to investigate the effects on amphibian species that are related to climate (i.e., the underlying secular movement), not interannual variability. This is why we use 30-year averages of model simulations for three time-horizons instead of averages over shorter periods of

time or annual data. As such, it does not allow to conclude anything about the effects of interannual or decadal variability on species' distributions. The bioclimatic variables used in this study represent climate and the ecological niche modelling approach used is static. Thus, it is very hard to infer how oscillations such as El Niño may impact on the geographical patterns of range contraction and expansions of individual species and the resulting geographical species richness loss. However, we consider these effects to be interesting and that they can be addressed in future studies about how climate and variability change could impact individual species and species richness patterns. This topic has been overlooked in the studies evaluating the impacts of climate change on species using niche modeling tools.

Minor comments

L. 52 "meridional"

Thank you. Typo has been corrected.

L. 63 This sentence is a little misleading, The reference 20 states about a weakening in the 70s due to the Great Salinity Anomaly, which was caused by coupled atmosphere/ocean dynamics in the Beaufort gyre, and caused an unprecedented AMOC anomaly in the 20th century only. Paleoproxy data shows that the AMOC may have not only weakened but collapsed during past deglacial events. Therefore, this statement has to be revised.

The sentence has been revised following the suggestions of the reviewer.

L. 67 Another reference could be added here on the CMIP6 models: Jourdain, N. C., Asay-Davis, X., Hattermann, T., Straneo, F., Seroussi, H., Little, C. M., and Nowicki, S.: A protocol for calculating basal melt rates in the ISMIP6 Antarctic ice sheet projections, The Cryosphere Discuss., <https://doi.org/10.5194/tc-2019-277>, in review, 2019.

The reference has been added.

L. 83 This sentence is disconnected and a little too general. The effects here are on NPP? Please specify.

Now the sentence is more specific in what the impacts could be.

L. 87 Could the authors state here how many bioclimatic indices are taken into account?

The number of bioclimatic variables has been added to the sentence.

L.96-97 From the box plots, it seems that there is not much variability from the extinction risk status either.

Thank you. The change has been made.

L.98 The variance between full dispersal is much lower than from no dispersal. Is there an explanation for this?

Thanks for the comment. These dispersal scenarios reflect the extreme responses of a species under climate change. Under a full dispersal, species can track their current climate requirements across the entire calibration area. By contrast, under a no dispersal, species cannot migrate outside their current distribution area. Accordingly, under a no dispersal scenario, species inevitably lose areas and therefore the variance will increase in the boxplots.

L. 106 Replace "as" by "such as".

Thank you, it has been replaced.

L.107-108 What is the explanation for the temperate regions to have lower loss?

Thanks for the comment. This explanation has been long debated in ecological literature. Some studies suggest that vulnerability to climate change is strongly linked with climate niche specialization (see Perez et al. 2016 Science 351(6280):1392-1393 for a more detailed explanation). In brief, this hypothesis states that temperate species as they have more generalist climate requirements than tropical species, they will be less more affected by future climate change impacts.

L. 318 Why is the BRT algorithm not listed here?

Corrected (see line 441).

L. 324 Is it the lowest omission rate or a low omission rate?

Low omission rate. Corrected (see line 446)

Figures:

Figure 2 - IPSL instead of ISPL

Thank you

Figure S11 - Missing numbers on the right colorbar.

Corrected

Figures S12 and S13 - These figures have low resolution. The lines and legend are very fuzzy. Colors for the Neotropical are different in the two figures.

Corrected

Figure S15 - I suggest changing the colorbar. Since the negative values are the ones important, and most of the values are negative, the colorbar range could cover only the negative range, and white for positives.

Corrected

References

- Anthoff D, Estrada F, Tol RSJ (2016) Shutting down the thermohaline circulation. *Am Econ Rev* 106:. <https://doi.org/10.1257/aer.p20161102>
- Araújo MB, Nogués-Bravo D, Diniz-Filho JAF, et al (2008) Quaternary climate changes explain diversity among reptiles and amphibians. *Ecography (Cop)* 31:8–15. <https://doi.org/10.1111/j.2007.0906-7590.05318.x>
- Arguez A, Vose RS (2010) The Definition of the Standard WMO Climate Normal: The Key to Deriving Alternative Climate Normals. *Bull Am Meteorol Soc* 92:699–704. <https://doi.org/10.1175/2010bams2955.1>
- Collins M, Sutherland M (2019) Extremes, Abrupt Changes and Managing Risks. In: Pörtner H-O, Roberts DC, Masson-Delmotte V, et al. (eds) IPCC Special Report on the Ocean and Cryosphere in a Changing Climate. In press, pp 3–63
- Defrance D, Ramstein G, Charbit S, et al (2017) Consequences of rapid ice sheet melting on the Sahelian population vulnerability. *Proc Natl Acad Sci U S A* 114:6533–6538. <https://doi.org/10.1073/pnas.1619358114>
- Deser C, Phillips AS, Alexander MA, Smoliak B V. (2014) Projecting North American climate over the next 50 years: Uncertainty due to internal variability. *J Clim* 27:2271–2296. <https://doi.org/10.1175/JCLI-D-13-00451.1>
- Dong B, Sutton RT (2007) Enhancement of ENSO variability by a weakened Atlantic thermohaline circulation in a coupled GCM. *J Clim* 20:4920–4939. <https://doi.org/10.1175/JCLI4284.1>
- Guisan A, Thuiller W, Zimmermann NE (2017) Habitat suitability and distribution models: With applications in R
- IPCC (2014) Climate Change 2014: Impacts, Adaptation and Vulnerability. Summary for Policy Makers. Cambridge University Press, Cambridge, United Kingdom and New York, NY, USA
- Jackson LC, Kahana R, Graham T, et al (2015) Global and European climate impacts of a slowdown of the AMOC in a high resolution GCM. *Clim Dyn* 45:3299–3316. <https://doi.org/10.1007/s00382-015-2540-2>
- Munguía M, Rahbek C, Rangel TF, et al (2012) Equilibrium of global amphibian species distributions with climate. *PLoS One* 7:. <https://doi.org/10.1371/journal.pone.0034420>
- Ochoa-Ochoa LM, Mejía-Domínguez NR, Velasco JA, et al (2019) Amphibian functional diversity is related to high annual precipitation and low precipitation seasonality in the New World. *Glob Ecol Biogeogr* 28:1219–1229. <https://doi.org/10.1111/geb.12926>
- Oliveira BF, Sheffers BR, Costa GC (2020) Decoupled erosion of amphibians' phylogenetic and functional diversity due to extinction. *Glob Ecol Biogeogr* 29:309–319. <https://doi.org/10.1111/geb.13031>
- Peterson AT, Soberón J, Pearson RG, et al (2011) Ecological Niches and Geographic Distributions (MPB-49). Princeton University Press

- Rocha JC, Peterson G, Bodin Ö, Levin S (2018) Cascading regime shifts within and across scales. *Science* (80-) 362:1379–1383. <https://doi.org/10.1126/science.aat7850>
- Schenk F, Välranta M, Muschitiello F, et al (2018) Warm summers during the Younger Dryas cold reversal. *Nat Commun* 9:1–13. <https://doi.org/10.1038/s41467-018-04071-5>
- Thuiller W, Guéguen M, Renaud J, et al (2019) Uncertainty in ensembles of global biodiversity scenarios. *Nat Commun* 10:. <https://doi.org/10.1038/s41467-019-09519-w>

Reply to Reviewer #2

Note: Original comments of the reviewer are included in italics, which is followed by our response in normal letter type.

We thank Reviewer #2 for her/his comments and suggestions which have helped us to greatly improve our manuscript.

My greatest criticism is that there is little discussion of how and why the changes impact amphibians. There is a mention in the methods about 5 bioclimatic variables, however there is a mention of 19 in the extended results. I assume the other 14 were not found to be important or something? There are some figures shown of two (annual mean temperature and precip) changes shown in the extended results, however the figure are so small and grainy that I can't make them out. Do you find that these two are the most important?

We have added a new section to the SI (section 2) to better explain the differences between the control and hosing simulations and describing the bioclimatic variables used. We have also clarified how the bioclimatic variables were selected (section S2.2). We detail how we have addressed these comments in our replies below.

1) The RCP8.5 and hosing scenarios can have different impacts on temperature and precipitation (ie RCP8.5 causes warming but AMOC weakening cools Europe, and effects on precipitation are different in different regions). I would like to see some figures and discussion linking the ecological impacts in different regions to the physical changes seen in those regions.

We appreciate this comment from the reviewer. To address the comment, we have extended the description and analysis of the climate scenarios used in our manuscript. The Supplementary information now includes a more complete explanation about the set of bioclimatic indices used in our manuscript and presents a more thorough analysis and comparison of the projections of such indices under the control and hosing experiments (see SI section 2). The differences between current climatic conditions and the future scenarios are described individually and in a multivariate context using Euclidean distances. We also included in the Methods section and the SI a discussion about the mechanistic links of these variables with demographic processes at local and regional scales for amphibian' species. For instance, we underline that the selected bioclimatic variables have a strong influence on species' distributions and functional traits (Araújo et al. 2008; Munguía et al. 2012; Ochoa-Ochoa et al. 2019; Thuiller et al. 2019) and that these bioclimatic variables have been used to model species' distributional areas both under current (Munguía et al. 2012) and future climate change conditions (Thuiller et al. 2019; Oliveira et al. 2020).

We also added Figure S30 in section 3.6 of the SI that illustrates the association at the grid scale of changes in climate and of impacts on amphibian species, as well as their spatial heterogeneity for the more severe hosing scenario (0.68 Sv) and the control scenario RCP8.5.

2) For instance I would expect less warming in the Arctic when the AMOC weakens (assuming that the hosing runs also include increasing GHGs) – does that mean less of an impact on the amphibians? Or are other factors more important?

In the new Section 2 of the SI we provide a more complete description and comparison of the climate scenarios used in our analysis. We also added some sentences to the main text explaining why, if the AMOC substantially weakens, climate conditions can be more dissimilar to current conditions and impacts larger. The reviewer is correct in that additional AMOC weakening could in principle be associated to lower impacts in the Palearctic region caused by lower warming levels in the high latitudes. However, in the hosing experiments, climate becomes considerably more dissimilar to current conditions than under the RCP8.5 scenario (Figure S16), which leads to higher impacts than under the RCP8.5 scenario (e.g., much larger contrasts between coldest and warmest months, large changes in precipitation). Paleoclimatic studies and model simulations suggest that under an AMOC collapse winters get much colder but that summers get considerably warmer in the Palearctic region (Schenk et al. 2018); this is also shown by the climate projections in our study (Figure S8). This is why the models project higher impacts for amphibian species in the hosing experiments in the Palearctic region. These climatic departures are illustrated in the new Figures S8-S16 that were added to the SI section 2.

In the new figures produced for the Palearctic realm (Fig. S14), which includes a portion of the Arctic region, we found a complex pattern of climate change involving notable changes in temperature and precipitation across the hosing experiments. For example, the annual mean temperature tends to decrease as the AMOC weakens in comparison with the control scenario. In contrast, the warmest month temperature (WMT) increases when the AMOC weakens, suggesting that summers might be hotter than in a control scenario not including this melting. These climate changes are captured by our ecological niche models as all species with Arctic distributions (*Salamandrella keyserlingii*, *Rana temporaria*, *Rana arvalis* and *Rana amurensis*) suffer notable range contractions under scenarios simulating an AMOC collapse (see Figure R1 below for a two examples).

Figure R1. Projections of potential distribution under climate change scenarios for two Palearctic frog species (*Rana temporaria* and *Rana arvalis*). a) Potential distribution (in red) for *Rana temporaria* under the RCP 8.5 control scenario and b) hosing experiment with sv 0.68 for 2030. c) Potential distribution (in red) for *Rana arvalis* under the RCP 8.5 control scenario and d) hosing experiment with sv 0.68 for 2030.

3) *Model description: Please add some more detail about the hosing experiments. From looking at the results I assume that the hosing experiments also have the time varying RCP forcing. Hence the effects of hosing are the differences between the hosing experiments and the RCP8.5 experiment. Please clarify. Also please add somewhere where the hosing is added and whether any compensation is applied.*

We have now clarified that the RCP8.5 is the baseline scenario to which different levels of additional freshwater are imposed. The RCP8.5 is used as the control scenario to which the hosing experiments are compared to. We have added model and experiment descriptions to the S1 section 2.1 in which we clarify the locations where the freshwater input was added:

“The climate simulations used in this study were produced with the *Institut Pierre Simon Laplace* low-resolution coupled ocean-atmosphere model (IPSL-CM5-LR)¹. The spatial resolution of the atmospheric component is 3.75°x1.875° in longitude and latitude, respectively, and includes 39 vertical levels. The nominal resolution of the oceanic component is 2° with a higher latitudinal resolution of 0.5° in the equatorial ocean, and 31 vertical levels. The locations for the release of the freshwater are deep water formation region in the North Atlantic (45°N to 65°N, 45°W to 5°E), which are classical regions of spread of the input of Greenland meltwater (Swingedouw et al. 2013). Since this freshwater input corresponds to Greenland ice sheet melting in terms of its long-term reservoir of freshwater, there is no reason to compensate for the water mass added, and no compensation is added in the model. Since this is a free surface model, this is not requested for conservation issues.”.

4) L23-24. *The authors talk about impacts being ‘enhanced by AMOC weakening’. There are a couple of points here. Firstly, the RCP8.5 scenario causes AMOC weakening as well as the hosing scenarios, so here and elsewhere I think you mean ‘additional AMOC weakening’. Also, the impacts on climate from an AMOC weakening alone are quite different from those from the increased greenhouse gases (GHG) in RCP8.5. Some impacts are of the opposite sign – ie increased GHG causes warming, but weakening the AMOC through hosing causes cooling over particular regions.*

Thanks for the comment. The reviewer is correct. The RCP8.5 includes some AMOC weakening, so we have clarified that additional AMOC weakening is superimposed on the RCP8.5 scenario. We corrected this in the main text.

Although the impacts do show opposite signs in the case of some species and pixels, the overall effect of additional AMOC weakening is negative in all regions for amphibians, in comparison with the control scenario (RCP8.5).

5) L27 and elsewhere. As someone outside the field, the terms for different regions are confusing and not defined. If these are common terms in this journal and for the readers then ignore this comment

The reviewer is right. Although the terminology is very familiar to biogeographers and ecologists, it could be new to a set of potential readers of this kind of work. We have included a map in the SI (section S1, Figure S1) that shows the delimitation of the biogeographical realms for readers outside the field. This kind of regionalization has been used for centuries by biologists starting with Alfred Russell Wallace and Charles Darwin. The terminology is very familiar to biogeographers and ecologists and we believe that with the help of Figure S1, the other readers will be able to easily get familiar to those terms and follow the paper.

6) L55-59. References used here are not really appropriate for the points made. Ref #18 is not looking at AMOC collapse and Ref #19 is not about impacts of a collapse at all. I suggest to refer to Ref 17, 24 and Jackson et al 2015

Thank you, the references have been changed.

7) L60-62. The main driver of AMOC weakening in future projections is actually changes in surface heat fluxes. See Gregory et al 2005. However there is large uncertainty about the L63. Reference #20 does not show this. Reference #17 (SROCC) is an alternative L64. Reference #22 doesn't seem to be used anywhere L67. Ref #23 does not show this. Ref #17 is an alternative

Thank you, the references have been changed.

8) L73-74. And the large uncertainty in AMOC response

Corrected.

9) L139-140. What do you mean by tipping points here? In what? Why do you think there are tipping points?

Thanks for the comment. We suggest that there is a possibility for an ecosystem tipping point because we found that, even with relatively small amounts of freshwater released, species distributions and the resulting geographical patterns of species richness changes in an abrupt (and possibly irreversible) manner. In other words, for relatively small additional weakening of AMOC, changes in the climate system will produce notable effects on biodiversity.

10) Fig 1. I don't think this figure is particularly interesting or necessary so I think it should be included in the supplementary material. As a climate scientist, it's unsurprising to see the AMOC weakening in all experiments, and I assume for biologists/ecologists, this doesn't mean a lot. The RCP8.5 scenario has increasing GHG (and the others may do too) so this has many impacts separate from the AMOC weakening. Instead it would be better to show the physical impacts that are affecting the ecology/biology. In particular I'd like to see more about the biometric indices

used. Possibly a map for each (of the most significant ones) at the end of the RCP8.5 scenario and the largest hosing scenario.

Thanks for the comment. We have sent Figure 1 to the SI as suggested by the reviewer. We have added a section (Section 2) of the SI in which we describe and discuss the changes in climate and included some maps about climatic dissimilarity metrics (see Figure S16). These multivariate metrics are used by ecologists working on modelling of climate change impacts. In addition, we have included maps for each of the bioclimatic variables used for each experiment as you suggested (see Figures S3-S7).

11) Fig 2. This is a horrible figure – it's really difficult to see whether there are any patterns and needs simplifying. I suggest that if there is no statistical difference across scenario or across family/realm/status then you don't show it all and note that you have looked in the text. That way you can concentrate on where there are differences. A clearer way of showing this data might be to have eg for each realm showing timeseries (three points with range) for each scenario. That way it is easier to compare the changes over time and scenario.

Thanks for the comment. We have modified this figure according to your suggestion of simplifying the plot; the patterns are much more readable now. We believe that the boxplots are the best way to summarize the variation and distribution shape of the response of species to climate change measured as a percentage of loss of current distributional area (i.e., range contraction). This type of figure (boxplot comparisons for different time-horizon, scenarios) are common in the ecology literature and this is why we prefer to report results in this way. This figure illustrates that the response is neither linear nor similar across biogeographical regions and extinction risk status (Figure 2a,b). We have put the figure 1c in the supplementary material to avoid confusions and make the figure more readable.

12) Fig 3. This is an interesting figure, but little discussed. Doing something like this for the biometric variables (see comments above) would help you investigate why there was a much greater loss in S Africa and why in the hosing experiments there was more loss in Europe and S/mid America.

We have included a section in the SI that focuses on describing the changes in bioclimatic variables under the different scenarios (see section 2 of SI). The new figures included there explore in which regions and bioclimatic variables the largest changes occur (see our reply to comment 2). Figure S30 shows the associations between climate departures and species loss for the 0.68sv and the RCP8.5 scenarios.

Please could you add something on one of the figures to show where the 6 biogeographical regions are for a wider audience.

Thanks for your suggestion. We have now included Figure S1 which shows in a map where the 6 biogeographical regions are located.

13) *Suppl L39. What are you normalising by for Euclidean distances? If you're using the average value in the control then that is a problem since values of some variables vary little compared to their absolute value (ie distance from zero) – for instance using temperatures in degrees C and Kelvin would give very different answers*

We normalised these variables due to high heterocedasticity and because we included variables from temperature (in °C) and precipitation (in mm); see (Garcia et al. 2014). The normalization is key for the calculation of Euclidean distances for each scenario and it is based on the average and standard deviation of bioclimatic variables from baseline climate (i.e., current climate). This makes quantities comparable. In this way, we can detect where non-analog climates emerges (i.e., the combination of multiple variables) and which are known to have a strong impact on species distributions (Williams and Jackson 2007).

Fig S12. This is way too small and when I zoom in is very grainy. I can't make out the different colours as well, probably because of the resolution. I suggest using less panels

Corrected

References

- Araújo MB, Nogués-Bravo D, Diniz-Filho JAF, et al (2008) Quaternary climate changes explain diversity among reptiles and amphibians. *Ecography (Cop)* 31:8–15. <https://doi.org/10.1111/j.2007.0906-7590.05318.x>
- Garcia RA, Cabeza M, Rahbek C, Araújo MB (2014) Multiple dimensions of climate change and their implications for biodiversity. *Science (80-.)*. 344
- Munguía M, Rahbek C, Rangel TF, et al (2012) Equilibrium of global amphibian species distributions with climate. *PLoS One* 7:. <https://doi.org/10.1371/journal.pone.0034420>
- Ochoa-Ochoa LM, Mejía-Domínguez NR, Velasco JA, et al (2019) Amphibian functional diversity is related to high annual precipitation and low precipitation seasonality in the New World. *Glob Ecol Biogeogr* 28:1219–1229. <https://doi.org/10.1111/geb.12926>
- Oliveira BF, Sheffers BR, Costa GC (2020) Decoupled erosion of amphibians' phylogenetic and functional diversity due to extinction. *Glob Ecol Biogeogr* 29:309–319. <https://doi.org/10.1111/geb.13031>
- Schenk F, Väliaranta M, Muschitiello F, et al (2018) Warm summers during the Younger Dryas cold reversal. *Nat Commun* 9:1–13. <https://doi.org/10.1038/s41467-018-04071-5>
- Swingedouw D, Rodehacke CB, Behrens E, et al (2013) Decadal fingerprints of freshwater discharge around Greenland in a multi-model ensemble. *Clim Dyn* 41:695–720. <https://doi.org/10.1007/s00382-012-1479-9>
- Thuiller W, Guéguen M, Renaud J, et al (2019) Uncertainty in ensembles of global biodiversity scenarios. *Nat Commun* 10:. <https://doi.org/10.1038/s41467-019-09519-w>
- Williams JW, Jackson ST (2007) Novel climates, no-analog communities, and ecological surprises. *Front Ecol Environ* 5:475–482. <https://doi.org/10.1890/070037>

Reply to Reviewer #3

Note: Original comments of the reviewer are included in italics, which is followed by our response in normal letter type.

1) Comment 1. Line 25 (Abstract), Line 132, and Line 162 – You described that large declines are projected for life-history traits. However, I did not find any information about life-history traits in the manuscript. Are you considering the IUCN extinction risk categories as life-history traits? If so, I would not recommend and suggest rewrite.

Thanks for the comment. We suggest that IUCN extinction risk categories can be a proxy for life-history traits because there are many evidences linking these categories with species' traits as geographical range, body size, reproduction (Reynolds 2003; Sodhi et al. 2008; Pearson et al. 2014; Chichorro et al. 2019). However, we clarify that this is a coarse proxy and more studies linking life-history traits with climate change driven extinction.

2) Comment 2. Lines 95-97 – Looking at the Figure 2B, I consider that under RPC8.5 scenario the projected range contractions are similar across extinction risk status. The distribution pattern of the medians is very similar to that of the high level taxonomic groupings. What criteria/analyses are you using to say that the projected range contractions for amphibians varies widely across biogeographical regions and extinction risk status but tend to be similar across high level taxonomic groupings?

Thanks for the comment. We have corrected this. In effect, the boxplots show that the medians and variance in range contraction varies only across regions. That variation is corroborated in the maps of percentage of amphibian species loss (Figure 2).

3) Comment 3. Lines 113-115 – I suggest that this sentence should be better elaborated. It is well-known that climate conditions under 2070 scenario will show more extreme climate conditions compared to previous ones. Considering that you are not taking into account the ability of extant species to adapt to ongoing climate change. It is not surprise that you find that range contractions tends to be higher toward 2070.

Thanks for the comment. We have rewritten this part of the text (see lines 111-113).

4) Comment 4. Lines 118-119 – Why the climate change scenario simulating a weakening of the AMOC reveal a contrasting pattern to that of the control RCP8.5 scenario? Looking at the results, my understanding is that range contractions are larger for simulations of AMOC than the control RCP8.5 scenario, but they are not contrasting pattern. Furthermore, you wrote (lines 125-126): “the range contraction of species is much larger compared to the control scenario across biogeographical realms ($F=407.1$, 127 $p<0.001$; Figure 2a), extinction risk status ($F=50.05$, $p<0.001$; Figure 2b), taxonomic groupings ($F=74.33$, $p<0.001$; Figure 2c), but not between freshwater discharge levels ($F=1.65$, $p=0.176$; Figure 2a-c)”. I do not know if you are writing about the three time horizons (2030, 2050, and 2070) or just 2030. Looking at the results (Figure 2a-c), in 2070 the simulations of AMOC and the control RCP8.5 scenario have similar range contractions for biogeographical realms, extinction risk status, and taxonomic groups.

Thanks for the comment. We have clarified that we refer only to 2030 and 2050. Around 2070, hosing scenarios simulating a large AMOC weakening will have similar projected individual range contractions but only under the no dispersal scenario. Under this scenario, species can colonize other sites outside their current distributional area. However, if you give the chance to a species to colonize (a full dispersal scenario) any pixel in each biogeographical realm, the range contractions will be more severe for AMOC weakening scenarios than for RCP 8.5 control scenario. This difference in dispersal scenarios is crucial to understand if species can be successfully in cope with these catastrophic climate-change scenarios. This is illustrated by the increase in variance toward values of 100% of range contraction. We have included this in the main text (see lines 123; 126-129; 149-152).

5) Comment 5, lines 166-168. This sentence is strange. Furthermore, the paper you are citing (Araujo et al. 2006, Journal of Biogeography) did not evaluate Neotropical region. I am curious because the Neartic and Palarctic regions have the lowest values for range contraction (Figure 2a), but have one of the highest species loss. You should explore these patterns between range contractions and species loss.

Thanks for the comment. First, we rewrote the sentence in the main text (see line 215) that refers to the Araujo paper (Araujo et al. 2006) and eliminated the comparison with tropical species. Also, we want to highlight that the Nearctic and Palearctic regions exhibit extensive range contractions under the climate change scenarios simulating the collapse of the AMOC but not under the RCP 8.5-only scenario (see Figure 1). These results contrast with the previous findings from Araujo et al. (2006) which suggest that future climate change would impact amphibian species but also could expand their current distributional areas.

Also, we want to clarify that we implemented two ways of measures climate changes on biodiversity: (i) at individual levels by calculating the percentage of range contraction for individual species and (ii) at assemblages level by calculating the percentage of species richness loss in each grid cell across geography (see lines 476-504 for further details). At the individual level, we calculated the percentage of range loss for only those species experiencing range contractions. A large proportion of modeled species in our study would exhibit range expansions in the future (see Table 2 for a full count). At the assemblage level, we calculated the percentage of species loss in each grid cell under each climate change scenario with respect to the number of current modeled species. Accordingly, if a region has a low number of species (e.g., Nearctic or Palearctic region) in comparison with others (e.g., Neotropical) and many species are projected to be lost in the future, the percentage of species loss will be higher in that region. We also noted that Nearctic and Palearctic regions exhibit both extensive individual range contractions and severe species loss in many sites. We agree with the Reviewer in that exploring these patterns in more detail would be interesting but we consider that this should be conducted in a new study incorporating several metrics of spatial turnover (i.e., beta diversity) for both species gaining and losing areas in the future.

6) Comment 6, Line 449, Figure 1 – Use “Atlantic meridional overturning circulation (AMOC) index at 26°N” instead of “AMOC index at 26°N”. Why are you not showing superimposed

hosing experiments that add 0.34-Sv of freshwater? Based on the legend I can't tell what the wider and thin lines represent.

Thanks for the comment. We have corrected the text, and we have now clarified that the thick and thin lines represent 30-year moving averages and annual frequency data, respectively. Moreover, as suggested by another Reviewer, we have sent this figure to the Supplementary Information.

7) Comment 7, Line 296-297 and Figure S1 – Did you project range contractions using resolution of 1° only for AMOC scenario A? What about other three scenarios?

Thanks for the comment. As the scenario A was very similar for 1° and 0.33°, we decided to perform the subsequent analyses using the resolution of 0.33°. We have clarified this in the text (see lines 477-480).

References

- Araújo MB, Thuiller W, Pearson RG (2006) Climate warming and the decline of amphibians and reptiles in Europe. *J Biogeogr* 33:1712–1728. <https://doi.org/10.1111/j.1365-2699.2006.01482.x>
- Chichorro F, Juslén A, Cardoso P (2019) A review of the relation between species traits and extinction risk. *Biol. Conserv.* 237:220–229
- Pearson RG, Stanton JC, Shoemaker KT, et al (2014) Life history and spatial traits predict extinction risk due to climate change. *Nat Clim Chang* 4:217–221. <https://doi.org/10.1038/nclimate2113>
- Reynolds J (2003) Life histories and extinction risk. In: Blackburn T, Gaston KJ (eds) *Macroecology*. Blackwell Publishing, Oxford, pp 195–217
- Sodhi NS, Bickford D, Diesmos AC, et al (2008) Measuring the Meltdown: Drivers of Global Amphibian Extinction and Decline. *PLoS One* 3:e1636. <https://doi.org/10.1371/journal.pone.0001636>

Reviewers' comments:

Reviewer #1 (Remarks to the Author):

This paper analyses the impacts of different levels of future AMOC slowdown on several amphibian species around the globe. The work is performed using model experiments of future climate change scenarios to force a niche model. Uncertainties relative to regional realms, extinction risk, modelling algorithm and ice sheet melting are taken into account.

The paper is better presented now, with the figures of higher quality. It is clear now that the 30 year averages would avoid the large interannual variability seen in the AMOC timeseries in Figure S2. The inclusion of Figures S3-S9 are a great addition to the paper, and explains some of the environmental drivers of the amphibian population changes. I read the manuscript and it believe is well presented, the figure changes were all included and are well reasoned.

I have a few minor comments that may improve the presentation of the figures in the Supplementary material. Otherwise, I think this paper will be a great contribution for the field and I recommend it to be published at Communications Biology.

Minor comments:

Supplementary:

L. 75 I think the 2 in CO₂ should be subscript not superscript.

Figure S3 - Why does this figure have 12 panels? Shouldn't follow the other figures with only 6 panels?

The Figures S3-S7: Although significant, these average changes seem to be small relative to the different scenarios. I wonder if the presentation could improve if the panels for the future scenarios were shown as anomalies to the current time or maybe the hosing scenarios were shown as anomalies to the RCP8.5.

Figures S10-S13 Caption should be corrected for precipitation seasonality (PS not PD).

Reviewer #3 (Remarks to the Author):

The authors have done a thorough job in reviewing the MS. However, I have two main comments that need to be looked at carefully to make sure the results are not skewed.

Comment 1: Looking at your figures S23-S29, the same niche algorithm (Maxent, BRT or CART), independently if the climatic scenario was RCP 8.5, A, B or C discharge experiment, showed equivalent results in 2030, 2050 and 2070. The hosing experiment D showed the same pattern, except for the BRT and CART algorithms in 2070 (Figure S27 f and i). These two exceptions are responsible for inflating the assemblage species loss in the Neotropical and Nearctic regions. The figure S29 clearly shows the point I am raising. The geographical patterns of species loss in 2030 for scenarios A, B, C and D (Figure S29 a,d,g and j) are equivalent. The same patterns is observed in 2050 for scenarios A, B, C and D (Figure S29 b,e,h and k). However, when we look at Figure S29-J indicating the species loss in 2070 for scenario D, the species loss is inflated to Neotropical and Nearctic regions. I was wondering if you are confident with the results of 2070 for the scenario D. I say that because your main results and discussion are based in these inflated results. If you compare the species loss between RCP 8.5 with those of hosing scenarios in 2030 and 2050, the hosing experiment still shows higher species loss than RCP 8.5, but this difference is not so discrepant as you are declaring. Furthermore, comparing only the results of the scenario D for 2030 and 2050, the Palearctic and Indomalayan would be the regions with larger species loss compared to RCP 8.5 scenario.

Comment 2: I am really intrigued with range contraction results showed in Figure 1 and those of species loss in Figure 2. For example, Figure 1 shows that amphibian species occurring in Afrotropical region will have their range reduced from 2030 to 2070 considering the 0.68-Sv freshwater scenario. However, the Figure 2 predicts that central parts of Africa will have species loss in 2030, but this same region will not present species loss in 2070 when the range contraction is more severe. Results of the Neotropical realm is even more intriguing. This region contains the highest amphibian species richness in the world with several species having small geographic ranges. Figure 1 shows that 50% of species occurring in Neotropical region will have more than 75% of their range reduced considering the RCP 8.5 scenario. However, Figure 2b indicates that this region will not be affected by species loss in 2070. That is intriguing. If assemblages in the Neotropical region are not losing species, it seems that the amphibian species are benefiting from climate change in this region. Could you include in Figure 2 the assemblages with species gain? Could you also provide a new figure like Figure 2 but considering no dispersal scenario? I think it you be great to see how dispersal is affecting these results.

Lines 27-28: It seems that you are referring only to the hosing experiments (A, B, C and D). It must be clear that independently of AMOC collapse, global warming is affecting climatically suitable areas by extensive range contraction (i.e. RCP 8.5).

Line 28. Delete "life history traits".

Line 104: You cited the Figure S2 in the line 100, and the Figures S18-19 in the line 104. I did not read the author guidelines, if the journal does not recommend that figures should be cited in the order they appear in the text, you can ignore this comment.

Reply to Reviewer #1

Note: Original comments of the reviewer are included in italics, which is followed by our response in normal letter type.

This paper analyses the impacts of different levels of future AMOC slowdown on several amphibian species around the globe. The work is performed using model experiments of future climate change scenarios to force a niche model. Uncertainties relative to regional realms, extinction risk, modelling algorithm and ice sheet melting are taken into account.

The paper is better presented now, with the figures of higher quality. It is clear now that the 30 year averages would avoid the large interannual variability seen in the AMOC timeseries in Figure S2. The inclusion of Figures S3-S9 are a great addition to the paper, and explains some of the environmental drivers of the amphibian population changes. I read the manuscript and it believe is well presented, the figure changes were all included and are well reasoned.

We are grateful for the constructive comments and suggestions provided by the reviewer.

I have a few minor comments that may improve the presentation of the figures in the Supplementary material. Otherwise, I think this paper will be a great contribution for the field and I recommend it to be published at Communications Biology.

Minor comments:
Supplementary:

L. 75 I think the 2 in CO₂ should be subscript not superscript.

Corrected.

Figure S3 - Why does this figure have 12 panels? Shouldn't follow the other figures with only 6 panels?

Corrected.

The Figures S3-S7: Although significant, these average changes seem to be small relative to the different scenarios. I wonder if the presentation could improve if the panels for the future scenarios were shown as anomalies to the current time or maybe the hosing scenarios were shown as anomalies to the RCP8.5.

We are grateful with the reviewer for this suggestion. The Figures S3-S7 were updated to show the anomalies of each hosing experiment to the RCP8.5. We split the maps to illustrate much better when warming/wettest or cooling/driest is occurring from each one of the hosing experiments to the RCP8.5. We believe that this presentation captures better the non-stationary of climate change in the hosing experiments and how much they differ with respect to the RCP8.5.

Figures S10-S13 Caption should be corrected for precipitation seasonality (PS not PD).

Corrected

Reply to Reviewer #3

Note: Original comments of the reviewer are included in italics, which is followed by our response in normal letter type.

The authors have done a thorough job in reviewing the MS. However, I have two main comments that need to be looked at carefully to make sure the results are not skewed.

We thank the reviewer for this comment.

Comment 1: Looking at your figures S23-S29, the same niche algorithm (Maxent, BRT or CART), independently if the climatic scenario was RCP 8.5, A, B or C discharge experiment, showed equivalent results in 2030, 2050 and 2070. The hosing experiment D showed the same pattern, except for the BRT and CART algorithms in 2070 (Figure S27 f and i). These two exceptions are responsible for inflating the assemblage species loss in the Neotropical and Nearctic regions. The figure S29 clearly shows the point I am raising. The geographical patterns of species loss in 2030 for scenarios A, B, C and D (Figure S29 a,d,g and j) are equivalent. The same patterns is observed in 2050 for scenarios A, B, C and D (Figure S29 b,e,h and k). However, when we look at Figure S29-J indicating the species loss in 2070 for scenario D, the species loss is inflated to Neotropical and Nearctic regions. I was wondering if you are confident with the results of 2070 for the scenario D. I say that because your main results and discussion are based in these inflated results. If you compare the species loss between RCP 8.5 with those of hosing scenarios in 2030 and 2050, the hosing experiment still shows higher species loss than RCP 8.5, but this difference is not so discrepant as you are declaring. Furthermore, comparing only the results of the scenario D for 2030 and 2050, the Palearctic and Indomalayan would be the regions with larger species loss compared to RCP 8.5 scenario.

We appreciate the comment from the reviewer about this. We are confident about the results for scenario D and 2070. The fact that all three niche modeling algorithms provide similar results for this scenario and horizon gives further evidence about the robustness of these results. We would consider the possibility of results being inflated if only one of the three algorithms produced such high impacts, but this is not the case. Moreover, these results support the argument about the existence of tipping point behavior in species losses. To further illustrate how much the additional reduction of AMOC affects amphibian species in comparison to the RCP8.5 scenario alone, we have calculated the difference in the geographical patterns of species losses (in percentage) between RCP8.5 and each one of the hosing experiments. The projected species loss in the hosing experiments reveals the highly non-linear response of the amphibian assemblages to these massive freshwater discharge scenarios and, as we mentioned in the main text, this suggest the possibility of a tipping point (see lines 193-196). We included three additional figures in the Supplementary Material to illustrate this point (Fig. S30-S32). Species losses driven by the hosing experiments tend to exhibit high values in several parts of the globe. In the hosing experiment with a strong freshwater discharge and for 2070 (Fig. S32) the species losses will be higher and more widespread in the Neotropical region. By contrast, the species losses driven by a high-emission scenario only (RCP8.5) are consistently concentrated in the same regions (e.g., south of Africa, some portions of Australia, North America). Accordingly, the difference between scenarios simulating a potential AMOC collapse and a high-emission scenario (RCP8.5) are notable and strongly heterogenous across geography.

Comment 2: I am really intrigued with range contraction results showed in Figure 1 and those of species loss in Figure 2. For example, Figure 1 shows that amphibian species occurring in Afrotropical

region will have their range reduced from 2030 to 2070 considering the 0.68-Sv freshwater scenario. However, the Figure 2 predicts that central parts of Africa will have species loss in 2030, but this same region will not present species loss in 2070 when the range contraction is more severe. Results of the Neotropical realm is even more intriguing. This region contains the highest amphibian species richness in the world with several species having small geographic ranges. Figure 1 shows that 50% of species occurring in Neotropical region will have more than 75% of their range reduced considering the RCP 8.5 scenario. However, Figure 2b indicates that this region will not be affected by species loss in 2070. That is intriguing. If assemblages in the Neotropical region are not losing species, it seems that the amphibian species are benefiting from climate change in this region. Could you include in Figure 2 the assemblages with species gain? Could you also provide a new figure like Figure 2 but considering no dispersal scenario? I think it you be great to see how dispersal is affecting these results.

We appreciate this comment from the reviewer. We have clarified in the legend of Figure 2 to avoid confusion. We consider that the two figures exhibit two different kinds of information about climate change impacts and therefore are not directly comparable. As we mentioned in the methods of the main text, we used two different approaches to analyze the effects of the climate scenarios we consider (starting in line 470). In the first approach, we calculated the traditional range shifts used in many modeling studies about the impacts of climate change on species. In this approach, we estimated how much of the current range will expand or contract in the future. In table S2, we provided information about the number of species that will contract and expand their ranges. These numbers show high variability across scenarios, time horizons, and niche modeling algorithms and range from 955 to 2068 species (Table S2). However, in Figure 1, we only included the species contracting their ranges. As the variation in the percentage of range expansions was very small and tends to be homogeneous across scenarios, we did not include this in the main text, but it is now included in the Supplementary Material (Fig. S37).

Figure 2, in contrast, was generated using an assemblage-based approach where modeled individual species were stacked for the current climate and for each scenario, time horizon, and niche modeling algorithm. Here, we included both species contracting and expanding their ranges. Then, we calculate the number of species co-occurring in each grid cell of $1^\circ \times 1^\circ$. Under this approach it is possible to evaluate the location of the amphibian species assemblages that will lose species in the future (i.e., changes in alpha diversity in ecological jargon).

This calculation is simply a subtraction of the future species richness minus current species richness. Positive values indicate gains in species numbers and negative values indicate losses in species numbers. Accordingly, it is not possible to compare the results from Figure 1 with Figure 2. We mentioned in the text the importance of using these two approaches to evaluate simultaneously which amphibian species and which amphibian assemblages (i.e., species pools) will be more sensitive to these impacts (lines 214-217 and 491-497 in the main text and section 3.6 in the Supplementary Material).

We have included the maps showing gains of species in amphibian assemblages in the Supplementary Material (Fig. S38-S39). Gains of species in regional assemblages are more frequent and pervasive in a high-emission scenario (RCP8.5) than in hosing experiments (Figure S38). Although the Neotropical region tends to exhibit some gains of species in comparison to other regions, the highest gains were toward temperate regions (e.g., the United States and Canada). Although these maps show the opposite patterns of species losses, these gains implicate that any species can colonize a given cell from any part of each realm and this can be unrealistic for many species due to limited dispersal. By contrast, species losses are more realistic because they are capturing local extinctions phenomena due to the

disappearance of climatic requirements for a given species in that site. We feel that a proper examination of these turnover species richness patterns requires further study incorporating metrics recently developed for temporal turnover (e.g., Magurran et al. 2019 *Global Ecol. Biogeog.* 28:1949-1960). We have included this discussion in the Supplementary Material (Section 3.6). We are grateful to the reviewer for highlighting this and we expect to explore this issue more comprehensively in another study.

Finally, under a no dispersal scenario, the patterns of species losses are consequently more drastic (Fig. S34-S36). The species losses under a high-emission scenario for the first time horizon are less severe than for the hosing scenarios (Fig. S34-a vs. Fig. S34-c). The differences are not very large across time horizons for experiment D (Figure S34-b vs. Fig. S34-d). In addition, the species losses were highly similar across hosing scenarios and time periods (Fig. S36). However, we consider that these no dispersal scenarios are likely unrealistic, because they reduce to zero the ability of species to track their climate requirements through geography. We feel that projected species losses under a no dispersal scenario can be extremely inflated. Although the amphibians tend to exhibit poor ability to disperse across space, some studies suggest that the amphibian responses to climate change might be idiosyncratic (e.g., Hickling et al. 2006 *Glob. Chang. Biol.* 12:450–455; Enriquez-Urzelai et al. 2019 *Clim. Change.* 154:289-301). We have included this discussion in the Supplementary Material (Section 3.6).

Lines 27-28: It seems that you are referring only to the hosing experiments (A, B, C and D). It must be clear that independently of AMOC collapse, global warming is affecting climatically suitable areas by extensive range contraction (i.e. RCP 8.5).

We have modified this part of the main paragraph as follows:

“Global warming impacts are severe and strongly enhanced by additional and substantial AMOC weakening, showing tipping point behavior for many amphibian species. Further declines in climatically suitable areas are projected across multiple clades, life history traits and biogeographical regions.”

We believe this clarifies the point raised by the Reviewer and let us stay within the word limits of this section.

Line 28. Delete “life history traits”.

Corrected

Line 104: You cited the Figure S2 in the line 100, and the Figures S18-19 in the line 104. I did not read the author guidelines, if the journal does not recommend that figures should be cited in the order they appear in the text, you can ignore this comment.

We prefer not changing the order of the figures in the Supplementary Information since it is a structured document in which the figures are part of additional discussion in structured sections and have an internal order within the SI.